# Cost-effectiveness analysis of PSA-based mass screening: Evidence from a randomised controlled trial combined with register data

Neill Booth[1]*, Pekka Rissanen[1], Teuvo L. J. Tammela[2,3], Paula Kujala[3,4], Ulf-Håkan Stenman[5], Kimmo Taari[6], Kirsi Talala[7], Anssi Auvinen[1]

1 Faculty of Social Sciences (Health Sciences), Tampere University, Tampere, Finland, 2 Department of Urology, Tampere University Hospital, Tampere, Finland, 3 Faculty of Medicine and Life Sciences, Tampere University, Tampere, Finland, 4 Department of Pathology, Fimlab Laboratories, Tampere, Finland, 5 Department of Clinical Chemistry and Haematology, University of Helsinki, Helsinki, Finland, 6 Department of Urology, University of Helsinki, Helsinki, Finland, 7 Finnish Cancer Registry, Helsinki, Finland

* Neill.Booth@tuni.fi

**Data Availability Statement:** Identifiable individual-level data cannot be shared publicly because of Finnish legislation governing the

## Abstract

In contrast to earlier studies which have used modelling to perform cost-effectiveness analysis, this study links data from a randomised controlled trial with register data from nationwide registries to reveal new evidence on costs, effectiveness, and cost-effectiveness of organised mass prostate-cancer screening based on prostate-specific antigen (PSA) testing. Cost-effectiveness analyses were conducted with individual-level data on health-care costs from comprehensive registers and register data on real-world effectiveness from the two arms of the Finnish Randomised Study of Screening for Prostate Cancer (FinRSPC), following 80,149 men from 1996 through 2015. The study examines cost-effectiveness in terms of overall mortality and, in addition, in terms of diagnosed men's mortality from prostate cancer and mortality with but not from prostate cancer. Neither arm of the FinRSPC was clearly more cost-effective in analysis in terms of overall mortality. Organised screening in the FinRSPC could be considered cost-effective in terms of deaths from prostate cancer: averting just over one death per 1000 men screened. However, even with an estimated incremental cost-effectiveness ratio of below 20,000€ per death avoided, this result should not be considered in isolation. This is because mass screening in this trial also resulted in increases in death with, but not from, prostate cancer: with over five additional deaths per 1000 men screened. Analysis of real-world data from the FinRSPC reveals new evidence of the comparative effectiveness of PSA-based screening after 20 years of follow-up, suggesting the possibility of higher mortality, as well as higher healthcare costs, for screening-arm men who have been diagnosed with prostate cancer but who do not die from it. These findings should be corroborated or contradicted by similar analyses using data from other trials, in order to reveal if more diagnosed men have also died in the screening arms of other trials of mass screening for prostate cancer.

protection of personal data. For data access requests, interested researchers can contact the relevant Finnish registries, for example, via https://thl.fi/en/web/thlfi-en/statistics/information-for-researchers/authorisation-application.

**Funding:** The study was supported by the Competitive State Research Financing of the Expert Responsibility area of Tampere University Hospital (grant numbers 9N064 and 9R002 to NB); the Yrjö Jahnsson Foundation (grant number 6572 to NB); the Academy of Finland (grant number 260931); the Pirkanmaa Cancer Society and the Cancer Society of Finland. The funding organisations did not play a role in the study design, data collection and analysis, decision to publish, or preparation of the manuscript and only provided financial support in the form of authors' salaries and/or research materials. Fimlab Laboratories provided support in the form of a salary for PK, but did not have any additional role in the study design, data collection and analysis, decision to publish, or preparation of the manuscript.

**Competing interests:** PK has taken part in a conference with support from Amgen. Fimlab Laboratories provided support in the form of a salary for PK, but did not have any additional role in the study design, data collection and analysis, decision to publish, or preparation of the manuscript. TT has acted as a consultant for Orion Pharma, Bayer AG, and Ferring and received research funding from Medivation, Pfizer, and Lidds AB. KT has taken part in a conference with support from Astellas and received research funding from Medivation, Astellas, and Pfizer. This does not alter our adherence to PLOS ONE policies on sharing data and materials.

# Introduction

There has been a wide range of evidence published on the effectiveness of systematic prostate-specific-antigen–based screening in reducing prostate-cancer mortality [1–4]; however, associated estimates of costs and cost-effectiveness from real-world data have received much less attention [5, 6]. Organised mass screening based on prostate-specific antigen (PSA) testing potentially offers systematic early detection of aggressive prostate cancer at a curable stage and thereby reduction of mortality [7]. However, the PSA test is not specific for cancer, as increased PSA levels can equally indicate benign changes in the prostate, so the PSA test has the potential to lead to harmful overtreatment [8]. Of course, questions extend beyond the clinical realm: policy-level ones can be asked, about what PSA-based organised screening might "cost" in relation to the "benefits" produced [9, 10]. Such relationships between costs and effectiveness (i.e., economic efficiency) are often described through some form of cost-effectiveness analysis (CEA) [11, 12]. While modelling-based CEA can provide useful information, its results typically are highly dependent on both the data and the assumptions used, which may sometimes be flawed or inaccurate [13, 14]. The need for assumptions can be minimised and the data quality maximised by drawing conclusions directly from the results of a pragmatic randomised controlled trial; we take this approach here, benefiting, e.g., from Finland's well established statutory health-care registries [15–17].

In light of the above considerations, a CEA was conducted with the primary aim of providing empirical estimates of some of the relationships between costs and effects of PSA screening from the Finnish Randomised Study of Screening for Prostate Cancer (FinRSPC) after 20 years of the trial, using intention-to-screen analysis of health-care costs, mortality, and cost-effectiveness.

# Materials and methods

## The FinRSPC

The complete age-based cohort for the FinRSPC was selected by staff at the Finnish population registry and consists of all men born in 1929–1944 residing in the Helsinki or Tampere region and alive on the date of randomisation (January 1 of each year from 1996 through 1999, a total of 80,458 men were randomised). Those men randomised to the screening arm were systematically invited for organised tests (serum PSA determination) at a local clinic, while those in the control arm received no such invitation as part of the trial. Three screening rounds were arranged, at four-year intervals, with men above 71 years of age no longer invited. Serum PSA was used for the primary screening test, with a cutoff of 4 ng/mL and ancillary testing for men with PSA 3.0–3.9 (digital rectal examination in 1996–1997, free/total PSA ratio from 1997 onwards). Randomisation occurred before consent, i.e., in order to prevent self-selection biases all men in the target age cohorts were randomised to one of the two arms without their consent being sought, this was undertaken in full accordance with Finnish legislation at that time. Follow-up started on January 1 in the year of randomisation and ended at death, upon emigration, or on the common closing dates for analyses of both costs and effectiveness (December 31, 2012–2015).

## Register-data permissions and sources

The collection of data for this research was approved by the relevant Institutional Review Boards: by the Finnish data-protection authority, by the National Institute for Health and Welfare (THL), by approval from Statistics Finland (TK-53-1330-18), and by the Ethics Committees of the participating university-hospital districts. The need for consent from the men

assigned to the trial was waived by a ruling from THL for the current register-based study (Official decision number: THL/36/5.05.00/2009). Data were obtained from several registries and entered in the FinRSPC database, using each man's unique Finnish personal identity code as the key for deterministic record linkage. Cancer cases were identified from the Finnish Cancer Registry (FCR), causes of death from Statistics Finland; episodes of hospital care from the THL-maintained Care Register for Health Care (CRHC), and prescription-medicine reimbursements from the nationwide register (PMRR) maintained by the Social Insurance Institution of Finland. The PMRR contains information on the exact costs of outpatient prescription medications paid by the healthcare sector in Finland. The CRHC is a comprehensive national register which covers inpatient stays in, and outpatient visits to, hospitals. To classify and identify resource use, we used the Finnish version of the Nordic Diagnosis Related Group (NordDRG) -system [18]. Identifiable individual-level data cannot be shared publicly because of Finnish legislation governing the protection of personal data. The data underlying the results presented in the study can be obtained from the relevant Finnish authorities for researchers who meet the criteria for access to confidential data. The funding organisations did not play a role in the study design, data collection and analysis, decision to publish, or preparation of the manuscript and only provided financial support in the form of authors' salaries and/or research materials. Fimlab Laboratories provided support in the form of a salary for PK, but did not have any additional role in the study design, data collection and analysis, decision to publish, or preparation of the manuscript.

## Costs

Our CEA follows a healthcare-sector perspective using register-based costs; utilising individual-level data on publicly-provided secondary and tertiary health-care visits and stays for men in the FinRSPC during the 20-year trial. In addition, the PMRR provides, to the nearest cent (¢), the costs of outpatient prescription medications paid by the Social Insurance Institution of Finland. For costs of secondary and tertiary care we used the most applicable NordDRG cost weights (in euros), which the THL had gathered from Finnish hospitals, for both inpatient and outpatient costs. The cost of the screening intervention itself was estimated by the FCR to cost approximately 50 euros per screen (including organisation of invitations, drawing of the blood sample, and PSA determinations but not any diagnostic evaluations, since the costs of diagnostic tests are captured in our other cost estimates). All results are rounded to the nearest 100 euros to yield a level of precision suitable for comparative estimates of cost and cost-effectiveness. Our base case analysis uses a discount rate of 3% per annum [19], and all euro amounts were adjusted using the most appropriate price indices available from Statistics Finland. Further details about these data sources and costs, as well as about the study design and trial registration have been provided in earlier FinRSPC or European Randomised Study of Screening for Prostate Cancer (ERSPC) publications ([3, 5, 20] or [2]).

## Analyses

The register data available on both costs and effects were analysed in accordance with the intention-to-screen principle; that is, they were examined in accordance with the initial trial-arm assignment. We used mortality as the measure of effectiveness, because no other register-based effectiveness data were available (e.g., on health-related quality of life) [20]. All follow-up is truncated at 17 years, with men who were randomised on January 1 1996 were followed up until December 2012, whereas, e.g., men who were randomised on January 1 1999 were followed up until December 2015. All tests of statistical significance are two sided, with Cox proportional hazards regression used in the mortality analysis. Our CEA calculates incremental

cost-effectiveness ratios (ICERs) by means of the data from the FinRSPC and national registers on costs and effects [21]. Our health-economic approach focuses on differences in mortality, including comparisons of the numbers of men dying between the two arms [19]. These health-economic comparisons are reported in line with current standards, with the primary result reported here being the overall ICER for the FinRSPC in terms of overall mortality [13], with additional CEA analysis for two subgroups, firstly for men who died from prostate cancer and, secondly, for men who died with, but not from prostate cancer [22]. All data handling and analysis, including the merging of data from different registers, was performed using Stata [23].

## Results

In all, 31,867 men were assigned to the screening arm and 48,282 to the control arm, with 3,788 men in the screening arm (11.9%) and 5,050 men in the control arm (10.5%) being diagnosed with prostate cancer, respectively (Fig 1 [24]). Data on both costs and effectiveness were recorded in the registers used in our study for 31,740 men in the screening arm (100%) and for 48,075 men in the control arm (100%). However, no cost records were found for 127 men in the screening arm and 207 men in the control arm, including one man in the screening arm who was diagnosed with and subsequently died of prostate cancer.

After 20 years of the trial, no statistically significant differences were observed between the arms in terms of the estimated average health-care costs of all men (Table 1). Although average costs for the 792 men who died of prostate cancer were around 10% higher in the screening arm (not statistically significant), there were negligible differences in total costs between the arms, since relatively few men died from prostate cancer in the screening arm. While average

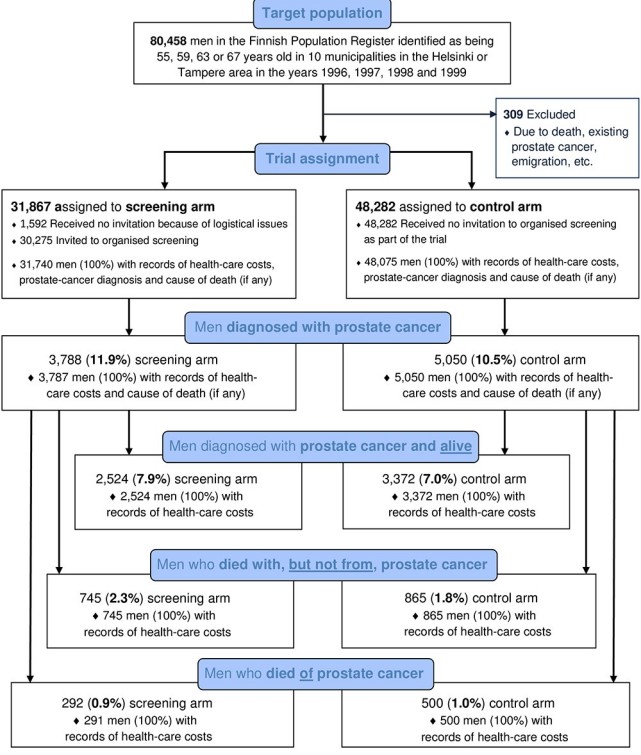

**Fig 1. Enrollment and health-related outcomes.**

**Table 1. Comparisons and statistical tests of the real-world health-care cost estimates.** Results comparing trial arms during the 17-year follow-up.

| Estimated all-cause health-care costs (register-based) | N in control arm | Mean in control arm | N in screening arm | Mean in screening arm | Difference between means (standard error) | Two-sided t-test | Difference[†] in total costs (in millions) |
|---|---|---|---|---|---|---|---|
| **All men** | 48,075 | **€37,800** | 31,740 | **€37,600** | -€200 (€400) | p = 0.65 | -€5.3 |
| Men **not diagnosed** with prostate cancer | 43,025 | **€36,100** | 27,953 | **€35,600** | -€500 (€400) | p = 0.26 | -€29.2 |
| Men **diagnosed** with prostate cancer | 5,050 | **€38,800** | 3,787 | **€39,300** | €400 (€1,100) | p = 0.64 | €23.9 |
| Men who have **survived with** a prostate cancer diagnosis | 3,372 | **€46,300** | 2,524 | **€46,700** | €300 (€1,100) | p = 0.76 | €14.7 |
| Men who **died with, but not from, prostate cancer** | 865 | **€62,400** | 745 | **€60,000** | -€2,400 (€3,100) | p = 0.43 | €9.7 |
| Men who have **died of prostate cancer** | 500 | **€63,600** | 291 | **€68,500** | €5,000 (€4,700) | p = 0.29 | -€0.5 |

[†] = Differences are calculated as total costs in screening arm minus total costs in control arm and adjusted to take account of the relative size of the trial arms (rounded to the nearest hundred thousand euros).

costs for the 1,610 men who died with but not from prostate cancer were approximately 5% lower in the screening arm (not statistically significant), a small substantive increase in total health-care costs for this subgroup was observed in the screening arm, as the rightmost column in Table 1 shows. This is because more men in this arm, i.e., a higher percentage of men in the screening arm, died with prostate cancer but not from it.

There was no statistically-significant difference in all-cause mortality (hazard ratio (HR) = 1.006, 95% confidence interval [CI], 0.98 to 1.03; P = 0.625) (Fig 2, Panel (A)). However, among diagnosed men there was a reduction in prostate-cancer-specific death in the screening arm: HR = 0.78, 95% CI, 0.68 to 0.90; P = 0.001 (Fig 2, Panel (B)). In addition, non-prostate-cancer mortality for men diagnosed with prostate cancer was higher in the screening arm than in the control arm: HR = 1.16, 95% CI, 1.05 to 1.27; P = 0.004 (Fig 2, Panel (C)). This increase in the rate of non-prostate-cancer mortality for men diagnosed with prostate cancer in the screening arm, seems to be most pronounced over five years after diagnosis (as Fig 2, Panel (C) shows). Overall, differences in mean health-care costs and mean effectiveness for diagnosed men between the trial arms were relatively small with regard to both prostate-cancer mortality and non-prostate-cancer mortality, with relatively high standard error (Table 2).

Our primary CEA produced a ICER which shows there was a health-related harm at less cost. This primary CEA result is presented in Fig 3, Panel (A), and shows that 95% confidence intervals are not able to be defined due to the uncertainty surrounding this estimate. Although not statistically significant, after 20 years, the impact of the FinRSPC equates to just under two additional deaths overall for every 1,000 men in the screening arm, with negligible savings in health-care costs. This finding reflects approximately 63 more deaths overall in the screening arm, which in turn reflects the negative contribution of approximately 136 more deaths observed in the screening arm for men diagnosed with prostate cancer, after having adjusted for the difference in size of the trial arms (Figs 1 and 2 and Table 2). Therefore, the ICER estimated for the FinRSPC overall, can be expressed qualitatively as a statistically non-significant reduction in costs accompanied by a statistically non-significant increase in the number of deaths (Table 2).

The estimated ICER for men diagnosed with, and who died of, prostate cancer, is 19,400€ per prostate-cancer death averted. This reflects the 38 or so fewer deaths from prostate cancer

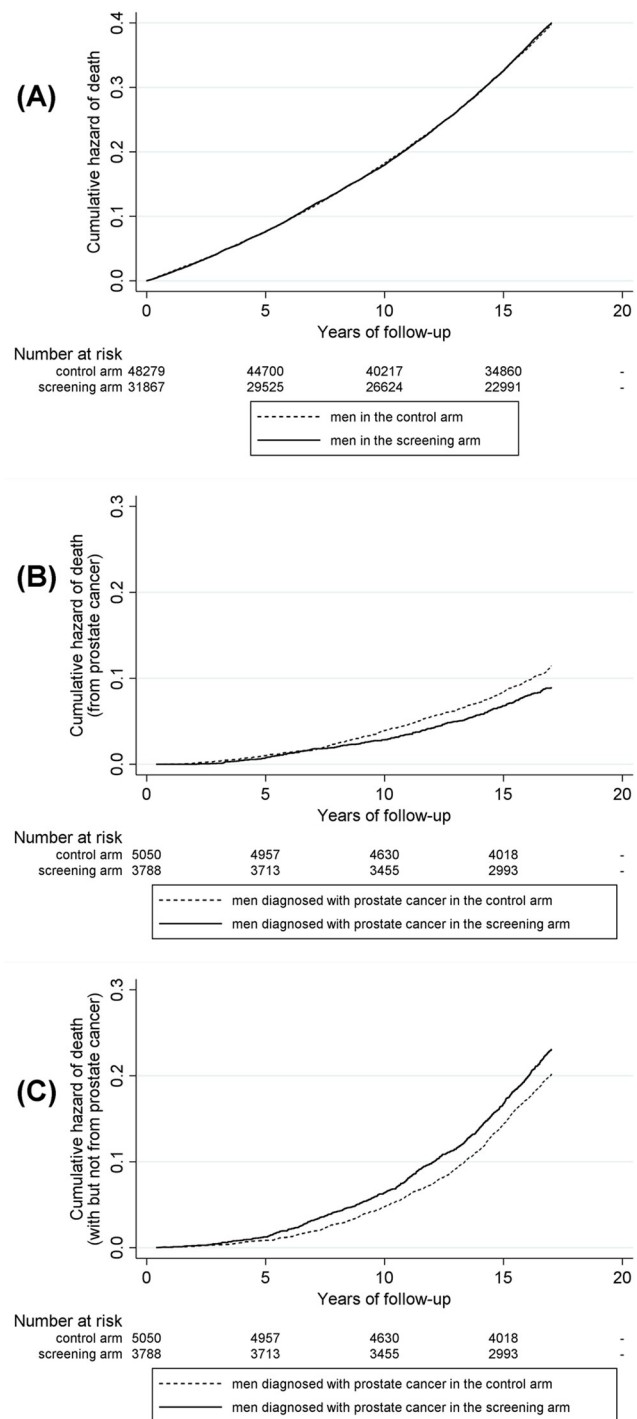

**Fig 2. Nelson–Aalen estimates of risk of dying, from point of randomisation. Panel (A):** Death from any cause, during follow-up, by trial arm. **Panel (B):** Death from prostate cancer, for men diagnosed during follow-up, by trial arm. **Panel (C):** Death with, but not from, prostate cancer, for men diagnosed during follow-up, by trial arm.

being observed in the screening arm after 20 years of the FinRSPC. However, only if a decision-maker's willingness to pay per 'prostate cancer death averted' is over 120,000€ could the screening arm of the FinRSPC be considered 'cost-effective' at conventional levels of statistical significance (Fig 3, Panel (B)).

**Table 2. Comparisons between the screening and control arms.** Register-based health-care cost estimates, observed effectiveness and incremental cost-effectiveness ratios, after 17 years of follow-up.

| | Control arm | | Screening arm | | Differences (screening arm—control arm) | | | Incremental cost-effectiveness ratio (ICER) |
|---|---|---|---|---|---|---|---|---|
| | Mean cost in euros | Mean effect (percentage of deaths*) | Mean cost in euros | Mean effect (percentage of deaths*) | in mean cost in euros | in mean effect (percentage of deaths†) | in number of deaths averted‡ | Point estimate [effectiveness measure] |
| **All men in the trial, using the effectiveness measure of deaths from any cause:** | | | | | | | | |
| Mean (total) | 37,800 | 0.327 | 37,600 | 0.329 | -100 | 0.001 | -63 | reduction in costs and increase in deaths§ [death from any cause] |
| (standard error) | (300) | (0.002) | (300) | (0.003) | (400)¶ | (0.001)¶ | | |
| **Men diagnosed with prostate cancer, using the effectiveness measure of deaths from prostate cancer:** | | | | | | | | |
| Mean (total) | 38,800 | 0.010 | 39,300 | 0.009 | 400 | -0.02 | 38 | 19,400€‖ [death from prostate cancer] |
| (standard error) | (600) | (<0.001) | (600) | (<0.001) | (900)¶ | (0.006) | | |
| **Men diagnosed with prostate cancer, using the effectiveness measure of cause of death something other than prostate cancer:** | | | | | | | | |
| Mean (total) | 38,800 | 0.018 | 39,300 | 0.023 | 400 | 0.02 | -174 | increase in costs and increase in deaths# [death with, but not from, prostate cancer] |
| (standard error) | (600) | (<0.001) | (600) | (<0.001) | (900)¶ | (0.008)¶ | | |

† = percentages expressed as decimals;

‡ = adjusted to take account of the relative size of the trial arms (rounded to the nearest integer);

§ = a 95% confidence interval is not able to be defined due to the uncertainty surrounding this estimate (see Fig 2, Panel (A));

¶ = bootstrap standard error;

‖ = an increase in mean costs (not statistically significant), and a statistically-significant increase in deaths averted (see Fig 2, Panel (B));

# = an increase in mean costs (not statistically significant) and a statistically-significant reduction in deaths averted, i.e., a statistically significant increase in deaths (see Fig 2, Panel (C)).

The analysis of cost-effectiveness above suggests a need to report on one further CEA too, this one focusing on death from other causes than prostate cancer among men diagnosed with prostate cancer (Fig 3, Panel (C)). This secondary analysis reports estimated cost-effectiveness for the men diagnosed with prostate cancer, 174 more of whom perished in the screening arm from causes other than prostate cancer, i.e., just over five additional deaths per 1000 men screened. Fig 3, Panel (C) also reflects the likelihood that these additional deaths also come at a cost in terms of health-care (of around 20,000€ per additional death).

## Discussion

We examined the costs, effectiveness, and cost-effectiveness connected with a large population-based comparative-effectiveness trial of organised PSA screening for prostate cancer. Taking each of these elements in turn, firstly, costs; in terms of mean health-care costs, we found no indications of statistically-significant differences overall. However, such differences may not be discernible due to the extensive heterogeneity observed in the trial participants' utilisation of health-care services; i.e., extremely high health-care costs for some men in the trial reduced the mean estimates' ability to fully describe the cost impact [5, 25]. For example, average and total overall costs for men not diagnosed with prostate cancer were, somewhat surprisingly, somewhat lower in the screening arm, even though men in the screening arm were attributed the additional cost of screening (Table 2). This may suggest differences in health-

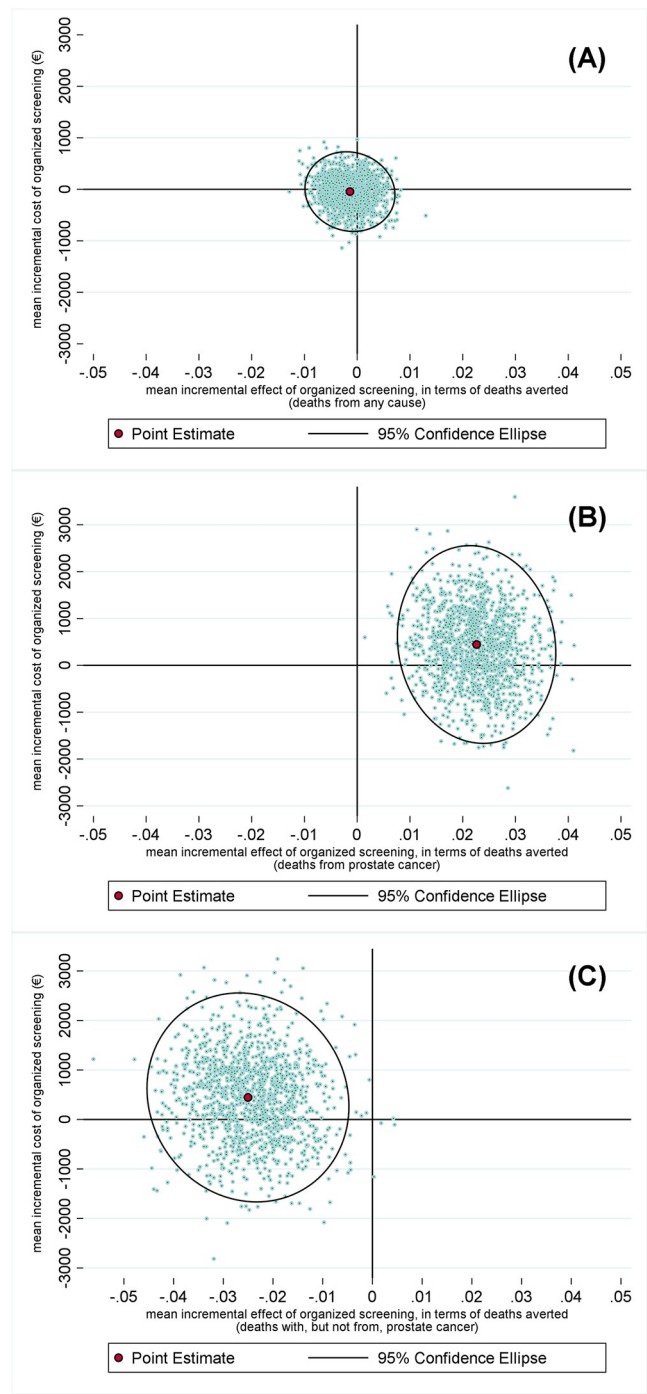

**Fig 3. Scatterplots of bootstrap replications of incremental cost-effectiveness ratios in terms of the number of deaths.** Panel (A): Estimates of incremental cost-effectiveness ratios in terms of death from any cause (for all men in the trial). Panel (B): Estimates of incremental cost-effectiveness ratios in terms of death from prostate cancer (for men diagnosed with prostate cancer). Panel (C): Estimates of incremental cost-effectiveness ratios in terms of death from causes other than prostate cancer (for men diagnosed with prostate cancer).

care costs in our study may have more to do with random fluctuations or outliers in health-care costs than screening itself. Of course, ideally the time horizon necessary for a comprehensive cost-effectiveness analysis would be one that is long enough for all relevant costs (and outcomes) to manifest (see, e.g., [26] or [19]). However, our analyses did not attempt to make predictions about future costs or survival for the trial population, as robust methods were not available to extrapolate from the health-care costs and mortality effects observed for men in the FinRSPC who have already died, to those who may die in the coming years [27]. One further reason for not extrapolating data beyond the within-trial horizon, is that the follow-up period of the trial covered an expanse of time that witnessed many changes in prostate-cancer treatment protocols.

Secondly, effectiveness, in exploratory mortality analyses prompted by our CEA findings in terms of all-cause mortality and prostate-cancer mortality, we also evaluated comparative effectiveness in terms of non-prostate-cancer mortality among the men diagnosed with prostate cancer. We undertook these secondary analyses because it was apparent from our other analysis results that, although on average prostate-cancer mortality was lower in the screening arm among men diagnosed with the disease, all-cause mortality was higher in the screening arm overall (even though this latter result was not statistically significant). To determine whether or not our CEA findings in terms of all-cause mortality were due to chance and given that the main impact of screening is, *a priori*, likely to be upon men diagnosed with prostate cancer, we undertook further analyses of mortality among men diagnosed with prostate cancer. The above secondary analyses are not, of course, undertaken in full accordance with the intention-to-screen (ITS) principle, however, three results reported in Fig 1, which do use the ITS principle, should be noted. The first result of note is that, relative to the control arm, on average 13% more men were diagnosed with prostate cancer in the screening arm (11.9% versus 10.5%). The second result of note is that, relative to the control arm, on average 13% less men died from prostate cancer in the screening arm (0.9% versus 1.0%). The third result of note in Fig 1 and, perhaps, the most important one here is that, relative to the control arm, on average 27% more men in the screening arm died with, but not from, prostate cancer (2.3% versus 1.8% in the control arm). A partial explanation for the result that, relative to the control arm, on average 27% more men in the screening arm died with, but not from, prostate cancer, could be related to overdiagnosis; with 13% more men on average diagnosed in the screening arm, relative to the control arm. Such a 'labelling' effect could plausibly account for approximately half of the observed additional non-prostate cancer deaths in diagnosed men. A second plausible explanation could be linked to competing causes of death; if more men are on average spared from death from prostate cancer due to PSA mass screening they may die of other causes. However, analysis using a proxy for survival time–i.e., the follow-up time in each arm–as the outcome measure (instead of number of deaths) also revealed an overall decrease in 'survival time' in the screening arm, for men diagnosed with prostate cancer. Put together, however, even the possible explanations listed above would still only seem to partially explain the finding of higher mortality for screening-arm men who have been diagnosed with prostate cancer but who do not die from it.

Although increased cardiovascular mortality due to endocrine therapy or the fear or stigmatization associated with cancer diagnosis may play some role in our findings about non-prostate-cancer mortality in men diagnosed with prostate cancer, such explanations remain only speculations as to why more men in the screening arm died with prostate cancer but not from it. In addition, we were not able to identify any single specific cause of death (or groups of causes of death), such as deaths related to intentional self-harm, or other underlying differences between the arms, which could explain these mortality differences. The most marked increases in non-prostate-cancer mortality were among those men in the screening arm with

Tumor-Node-Metastasis -stage T1c cancers at diagnosis (i.e., impalpable cancers detectable only by PSA testing [28]). Although our data suggest mid-level socioeconomic status may have been associated with an increase in non-prostate-cancer mortality when men were diagnosed with prostate cancer at any other than stage than stage T1c, these findings were not statistically significant. Possibly due to the relatively small number of observed deaths at this stage, such adjustments for socioeconomic status had minimal material impact on the differences between the arms in terms of mortality among men with prostate cancer. Further explanatory analysis is beyond the scope of this exploratory study.

Thirdly, we turn to the results relating to the cost-effectiveness of organised screening in the FinRSPC, which varied according to the outcome measure used. We present three ICERs: there was negligible impact of mass screening in the FinRSPC in terms of death from any cause in all men, (what can be interpreted as) a positive impact for death from prostate cancer in diagnosed men, and (what can be interpreted as) a negative impact for death with, but not from, prostate cancer in diagnosed men. Sensitivity analysis showed that using a discount rate of 5% and 1% does not result in major changes in the differences in costs or cost-effectiveness between the two arms in any of these analyses. In their assessment of the cost-effectiveness of screening, epidemiological studies have focused mainly on disease-specific mortality [2, 29, 30], often to the exclusion of any other effects on mortality [31]. In contrast to earlier CEAs [32–36] our approach to health-economic evaluation considers not merely prostate-cancer mortality; but characterizes all-cause mortality too, along with non-prostate-cancer mortality in men diagnosed with prostate cancer. One potential pitfall in modelling cost-effectiveness in a manner which does not adequately question the underlying epidemiology is that, accordingly, any errors in the choice of outcome measures may be compounded in the act of modelling. If the choice of outcome measure is restricted by epidemiological convention this may obscure relevant effects of the intervention, resulting in models neglecting to include a potentially relevant health state, such as non-prostate-cancer mortality in men diagnosed with prostate cancer. Incorporating all potentially important mortality impacts should be seen as central in any health-economic evaluation [37].

The interpretation of incremental cost-effectiveness ratios is not a straightforward matter, as the process of interpretation is typically specific to both the ICER's content and the decision-making context in question. In the field of health-economic evaluation, a cost-effectiveness ratio usually represents some indicator of the amount of health *gained* divided by some estimate of the financial costs associated with that estimated 'health' *gain*. Only relatively infrequently, as in the secondary analysis presented here, does the cost-effectiveness ratio represent some indicator of the amount of health *lost* divided by some estimate of the financial costs of that estimated 'health' *lost*. When we report that, for men diagnosed with prostate cancer, the estimated health-care cost per additional death is around 20,000€, this means that the data suggests the trial was economically efficient at increasing non-prostate-cancer mortality for those men. It should be clear from the results presented here that cost-effectiveness ratios can contain or omit a wide range of factors. For this reason, understanding the content of each incremental cost-effectiveness ratio is important when they are interpreted, for example, how well costs and health effects are measured and analysed, and what costs and effects are, or are not, included in the analysis. In Table 3 we set out the main research assumptions and key components which underpin the health-economic evaluation of the FinRSPC.

In practice, interpretation of cost-effectiveness information requires understanding of the components and qualities of that information [39]. Table 3 is intended to provide a useful starting point for interpretation of the information about costs, effects and cost-effectiveness provided by our study [40]. Interpretation of CEA results is also usually influenced by the interplay between the decision-making context and the specific information provided by the

**Table 3. Main assumptions.** Key elements of the health-economic evaluation of the FinRSPC.

| | |
|---|---|
| **A) Key elements of the analysis related to the FinRSPC:** | |
| strengths | • Assignment to the trial arms occurred without prior consent (but with the permission of the authorities)<br>• The men randomised represented the whole target population of the Tampere and Helsinki areas during the period, i.e., all registered male citizens of the selected age groups were included<br>• Long-term register-based follow-up was available for practically all men (>99.9%) |
| limitations | • Neither Finnish registries or the trial database includes consistent follow-up of either many of the possible health-related impacts, or some of the costs, associated with prostate-cancer screening<br>• The FinRSPC is limited by its context, e.g.:<br> • clinical practice today may be quite different to that of the late 1990s<br> • PSA testing became more prevalent in the population over the period of the trial, which seems likely to have had a significant effect on the impacts of the screening intervention [38]<br>• The long duration of follow-up may also mean that more influences unrelated to the screening trial are reflected in its results, i.e., that there is more 'noise' in the data<br>• Clinical trials such as the FinRSPC typically can only provide robust information on average treatment effects for the whole trial population. This is also the case for this trial, which practically precludes robust analysis by, e.g., geographical- or age-related–subgroup |
| **B) Key elements of the analysis related to costs:** | |
| strengths | • The analysis uses well-established registers covering both use of hospital services (inpatient and outpatient) as well as reimbursements for almost all outpatient prescription medications<br>• The registers provide almost complete coverage of these (hospital and prescription-medication) costs for almost all men in the trial for almost the whole duration of the 20-year study |
| limitations | • In principle, ideally all costs associated with PSA mass screening for prostate cancer and its consequences might be included as part of a cost-effectiveness analysis, at least when attempting to gauge the robustness of the results to the inclusion or omission of a range of cost items.<br>• Although information relating to primary care costs is typically included, such information was not readily available from Finnish registers or the trial database, so is not included here<br>• Various cost drivers, such as costs to patients, costs which fall on the social-care budget, and costs of lost productivity in the economy, were not included in our analyses<br>• Although the registers provide an identical source of data for men in both arms of the trial, and although price indices and discount rates were applied uniformly in both arms, the register-based cost estimates presented here are based on NordDRG cost weights, the cost estimates are, at best, merely rough indicators of the magnitude of the true current costs which might be associated with PSA mass screening for prostate cancer |
| **C) Key elements of the analysis related to health-related outcomes:** | |
| strengths | • The analysis presented here focuses on one of the most important and robust impacts related to health outcomes, i.e., mortality<br>• The analysis uses data from well-established registries and precise cause-of-death registers with practically complete coverage (at least for men who did not emigrate) |
| limitations | • No direct measurement of health-related quality of life or patient satisfaction was possible using the available register data |
| **D) Key elements related to the cost-effectiveness analysis *per se*:** | |
| strengths | • Each incremental cost-effectiveness ratio (ICER) presented looks at a different aspect of mortality, together the three ICERs presented provide an a variety of useful indicators of the efficiency of mass screening for prostate cancer in terms of the main impacts on mortality<br>• Although the cost drivers used in our analysis are limited in scope, as noted above (in section **B**) of the table), the data provides almost complete coverage of two main costs: hospital care and prescription medications |
| limitations | • Each incremental cost-effectiveness ratio (ICER) presented looks at a different aspect of mortality, none of the ICERs alone provide an all-encompassing indicator of the efficiency of organised PSA mass screening for prostate cancer<br>• As noted above (in section **C**) of this table) our analysis does not incorporate health-related quality of life considerations or considerations relating to patient satisfaction. For this reason the ICER estimates presented here provide only a truncated representation of the efficiency of PSA mass screening for prostate cancer and do not take into account important effects, e.g., on quality of life |

incremental cost-effectiveness ratios in question. Therefore, judgment will typically be needed, in every separate case and context, to gauge to what extent any estimated incremental cost-effectiveness ratios provide an indication of 'value'. Many elements of the chosen approach to health-economic evaluation can markedly influence the results of cost-effectiveness analyses [41]. When CEAs are based on a single randomised controlled trial, CEAs naturally are heavily dependent on that source of information. Although randomised controlled trials are typically seen as one of the best research methods to inform public health policy, it should be noted that they do have their weaknesses [25, 42, 43]. Further details about the strengths and limitations of our study will be set out below.

## Strengths of the study

Our register-based cost-effectiveness analysis combines the power of a randomised controlled trial with extensive follow-up via real-world data from comprehensive health-care registers. By avoiding reliance on many of the assumptions typically necessary for modelling costs and outcomes, our study represents a potentially significant application of CEA to improve the knowledge base about organised screening for prostate cancer. Although numerous modelling-based studies have been reported upon [32–36, 44], their estimates or forecasts typically do not proceed from data alone, with a frequent cascading effect wherein cost estimates are based on previous estimates of outcomes. Our results can be regarded as a groundbreaking contrast, in that this is the first report on CEA based on real-world data derived from one study of PSA mass screening. The men in the FinRSPC, i.e., in the Finnish arm of the ERSPC, were a complete age cohort of the men in and around two main Finnish conurbations, Helsinki and Tampere. The men were assigned to the two arms before randomisation, thus minimising problems associated with selection to either group. Although the FinRSPC does not provide a perfectly valid assessment of organised screening versus no screening, it likely provides a potentially valid assessment of organised screening versus current clinical practice. The contamination by opportunistic PSA-testing experienced during this trial is more likely to be generalizable to current clinical practice than would 'no screening', by providing evidence of the likely impact of organised screening over and above opportunistic testing [19]. In addition, the data over the 20 years of the trial (with 17-year median follow-up time), from fairly comprehensive data on health-care costs and on effectiveness (in terms of mortality), help in obtaining potentially generalizable cost-effectiveness estimates, which realistically account for the diluting effects of contamination.

Truncation of the follow-up at 17 years was undertaken to limit our analysis to only the most robust data, because as age cohorts were selected from the population the on January 1 of each year from 1996 through 1999, analysis without truncation would mean that follow-up beyond 17 years would only be possible for fewer and fewer men each year until the maximum of 20 years of follow-up. Truncation in this study produces more conservative results, with analysis using all available data producing both larger effect sizes and more statistically-significant associations.

Another strength of our study lies in its ability to inform current and future choices of suitable metrics for effectiveness [45]. For instance, our finding that over the 20 years of the FinRSPC trial, mortality from causes other than prostate cancer among diagnosed men increased in the screening arm has potentially significant implications for future research. Our analysis likely provides a useful building block in that its findings could be input for testing existing cost-effectiveness models' sensitivity to new information, which has been shown to be useful elsewhere [46]. In addition, similar analyses from comparable trials of organised screening could provide illuminating corroboration or contradiction of the findings presented here,

because we are not aware of any other published analyses of mortality from causes other than prostate cancer among diagnosed men in trials of PSA mass screening for prostate cancer [47].

## Limitations of the study

As set out in parts of Table 3, this study is limited in scope for a number of reasons, e.g., that no direct analysis of health-related quality of life or patient satisfaction was possible using the available register data [48]. The estimates from a pragmatic trial in Finland presented here are not necessarily indicative or representative of the impacts of organised PSA screening likely in other health-care systems. Indeed, these estimates are unlikely to represent exact health-care costs, effectiveness (in terms of mortality), or cost-effectiveness in other settings. However, this is unavoidable for any pragmatic long-term, real-world study. The generalizability of our analysis to other settings is dependent on how well the manner of implementing the screening intervention and subsequent care pathways in the FinRSPC can be generalized and also on contextual elements such as the treatment patterns and the relative homogeneity of the FinRSPC participants (the vast majority being Finnish and Caucasian). More generally, the health-care system in which the trial took place (in the largely publicly-funded Finnish health-care system) may limit generalizability. Despite these limitations, this trial of comparative effectiveness does represent an important source of evidence, which can be used to supplement earlier evidence from modelling studies. Of course, the authors acknowledge that, just as modelling-based CEA depends on assumptions that may sometimes be flawed or inaccurate, the relevance of empirical CEA to a wider population or time horizon also depends on assumptions that may sometimes be flawed or inaccurate. In addition, all our results should be interpreted in consideration of the likelihood of high levels of contamination in the control arm [49], since most of the men in the control arm had a PSA test at some point in the trial and the cumulative incidence of T1c cancers was, for example, only approximately 20% higher in the screening arm than in the control arm. Although almost 75% of men in the screening arm of the FinRSPC participated in the organised screening, we cannot know exactly which men underwent non-systematic screening, i.e., opportunistic testing [38]. Further, the results presented here for the subgroups of men diagnosed with prostate cancer are not necessarily causally linked to randomisation, they are the result of randomisation followed by diagnosis, so the intention-to-screen-analytic comparison between the arms is uncertain in this respect [25]. However, the results of the intention-to-screen analysis show mortality effects of similar magnitude to those in the subgroup analysis presented here. It should also be noted here that health-economic evaluations are information-intensive in their input requirements and that their use often suffers from a lack of appropriate information [50], especially as pathways to health are quite complex [51]. For example, the main costs analysed were from secondary and tertiary care, so various cost drivers, such as costs of lost productivity to the economy, costs of primary-care treatment, and costs due to social care, were not considered. One of these, primary-care costs, were not included in our study as data were not available from registers for the majority of the follow-up period (except to the extent that prescription medication use, as part of primary care, was covered). However, we are not aware of any strong reason why primary care costs would differ substantially between the groups when, e.g., there were no major differences in the costs of secondary or tertiary care. One further limitation (or, conversely, potential strength), is that our study employed a fixed time horizon, setting it apart from many model-based studies, which vary the time horizon modelled. On the other hand, models that attempt to estimate "lifetime" costs and effectiveness typically rely on assumptions that could seem out of place in light of the findings presented here, especially since we cannot know with any certainty which direction the impacts of PSA screening will take next.

## Conclusion

Our primary analyses showed no major difference in overall health-care costs or in overall mortality within the 17 years of follow-up. However, in further analysis, relatively minor reductions in prostate-cancer mortality at the expense of increased costs in the screening arm were found, but these may be outweighed by an increase in mortality from other causes for men diagnosed with prostate cancer in the screening arm. Our analysis could be usefully supported or contradicted by similar analyses using data from comparable trials of mass screening. Longer-term follow-up may also allow more robust conclusions as to the balance of the benefits and harms of introducing organised PSA mass screening.

## Acknowledgments

The authors would like to express thanks to all those who have helped in collating the trial data over the past 20 years but especially to Liisa Määttänen, formerly of the Finnish Mass Screening Registry, for her continued support related to the trial database, even in retirement. The corresponding author would also like to thank Jani Raitanen and Pasi Aronen for numerous helpful discussions about matters statistical and analytical. Any mistakes that may remain are the responsibility of the corresponding author.

## Author Contributions

**Conceptualization:** Neill Booth, Pekka Rissanen, Teuvo L. J. Tammela, Anssi Auvinen.

**Data curation:** Neill Booth, Ulf-Håkan Stenman, Kirsi Talala.

**Formal analysis:** Neill Booth, Anssi Auvinen.

**Funding acquisition:** Neill Booth, Pekka Rissanen, Teuvo L. J. Tammela, Kimmo Taari, Anssi Auvinen.

**Investigation:** Neill Booth, Paula Kujala, Kimmo Taari, Kirsi Talala.

**Methodology:** Neill Booth, Pekka Rissanen, Paula Kujala, Anssi Auvinen.

**Project administration:** Neill Booth, Kirsi Talala.

**Supervision:** Pekka Rissanen, Anssi Auvinen.

**Validation:** Neill Booth, Anssi Auvinen.

**Visualization:** Neill Booth.

**Writing – original draft:** Neill Booth, Anssi Auvinen.

**Writing – review & editing:** Neill Booth, Pekka Rissanen, Teuvo L. J. Tammela, Paula Kujala, Ulf-Håkan Stenman, Kimmo Taari, Kirsi Talala, Anssi Auvinen.

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
