## [Decision Letter · Decision Letter 0]

26 Sep 2019

PONE-D-19-24756

Register-based cost-effectiveness analysis of organized prostate-specific antigen screening: Evidence from a randomized controlled trial

PLOS ONE

Dear Mr. Booth,

Thank you for submitting your manuscript to PLOS ONE. After careful consideration, we feel that it has merit but does not fully meet PLOS ONE’s publication criteria as it currently stands. Therefore, we invite you to submit a revised version of the manuscript that addresses the points raised during the review process.

We would appreciate receiving your revised manuscript by Nov 10 2019 11:59PM. To enhance the reproducibility of your results, we recommend that if applicable you deposit your laboratory protocols in protocols.io, where a protocol can be assigned its own identifier (DOI) such that it can be cited independently in the future. For instructions see: http://journals.plos.org/plosone/s/submission-guidelines#loc-laboratory-protocols

We look forward to receiving your revised manuscript.

Kind regards,

Christopher J.D. Wallis, MD, PhD

Academic Editor

PLOS ONE

Journal Requirements:

1. Please provide additional details regarding participant consent. In the ethics statement in the Methods and online submission information, please ensure that you have specified (1) whether informed consent was obtained after randomization and (2) what type you obtained (for instance, written or verbal). If your study included minors, state whether you obtained consent from parents or guardians. If the need for consent was waived by the ethics committee, please include this information.

We also ask that you please provide the full names of all of the institutions or hospitals from which you obtained ethics approval.

2. Thank you for including your competing interests statement; "Paula Kujala has taken part in a conference with support from Amgen. Teuvo Tammela has acted as a consultant for Orion Pharma, Bayer AG, and Ferring and received research funding from Medivation, Pfizer, and Lidds AB. Kimmo Taari has taken part in a conference with support from Astellas and received research funding from Medivation, Astellas, and Pfizer. None of the other authors have anything to disclose. The authors have declared that no competing interests exist."

4. Please include a caption for figure 1

Reviewers' comments:

Reviewer's Responses to Questions

**Comments to the Author**

1. Is the manuscript technically sound, and do the data support the conclusions?

Reviewer #1: Yes

2. Has the statistical analysis been performed appropriately and rigorously? 

Reviewer #1: Yes

3. Have the authors made all data underlying the findings in their manuscript fully available?

Reviewer #1: No

4. Is the manuscript presented in an intelligible fashion and written in standard English?

Reviewer #1: Yes

5. Review Comments to the Author

Reviewer #1: Booth et al. examined cost-effectiveness in terms of overall mortality and prostate cancer mortality among men included in the FinRSPC prostate cancer screening trial. Neither the screening or control arm was overall cost-effective. Organized screening could be considered cost-effective in terms of prostate cancer death by preventing one death per 1000 men screened.

This is a well-written manuscript with appropriate methodology and limitations listed. A few minor recommendations for improving the manuscript:

1) Why was all follow-up truncated at 17 years? The authors state that they did it, but did not provide further rationale.

2) It would be helpful to have a Table (supplementary) or two listing assumptions, costs, etc

6. PLOS authors have the option to publish the peer review history of their article (what does this mean?). If published, this will include your full peer review and any attached files.

Reviewer #1: No

---

## [Author Response · Author response to Decision Letter 0]

11 Oct 2019

Responses to editor comments:

1) We have added the following text to the methods section and in the online submission system: 

”The need for consent from the men assigned to the trial was waived by a ruling from THL for the current register-based study (Official decision number: THL/36/5.05.00/2009)” 

2) We have amended the Funding and Competing Interests statements in line with your requests and in line with those sent by Vicky Stabler in e-mail correspondence on the 27th of September, as follows:

”The funding organisations did not play a role in the study design, data collection and analysis, decision to publish, or preparation of the manuscript and only provided financial support in the form of authors' salaries and/or research materials. Fimlab Laboratories provided support in the form of a salary for PK, but did not have any additional role in the study design, data collection and analysis, decision to publish, or preparation of the manuscript.”

3) Identifiable individual-level data cannot be shared publicly because of Finnish legislation governing the protection of personal data. The data underlying the results presented in the study can be obtained from the relevant Finnish authorities for researchers who meet the criteria for access to confidential data. We have added text to this effect in the manuscript.

4) Thank you for noticing this omission, we have added: “Figure 1: Enrolment and health-related outcomes.”

Responses to reviewer comments:

5) We thank the reviewer for their constructive comments!

1. In response to the comment about why we analysed truncated data, we have added the following text: 

”Truncation of the follow-up at 17 years was undertaken to limit our analysis to only the most robust data, because as age cohorts were selected from the population the on January 1 of each year from 1996 through 1999, analysis without truncation would mean that follow-up beyond 17 years would only be possible for fewer and fewer men each year until the maximum of 20 years of follow-up. Truncation in this study produces more conservative results, with analysis using all available data producing both larger effect sizes and more statistically-significant associations.”.

2. We have implemented the useful suggestion of a table listing the assumptions, costs, etc. which underpin our findings. We have added a table and related supporting text in order to improve the discussion section and facilitate understanding of the results presented in our manuscript.

---

## [Editor Report · Decision Letter 1]

16 Oct 2019

Register-based cost-effectiveness analysis of organised prostate-specific antigen screening: Evidence from a randomised controlled trial

PONE-D-19-24756R1

Dear Dr. Booth,

We are pleased to inform you that your manuscript has been judged scientifically suitable for publication and will be formally accepted for publication once it complies with all outstanding technical requirements.

With kind regards,

Christopher J.D. Wallis, MD, PhD

Academic Editor

PLOS ONE
---

## [Editor Report · Acceptance letter]

25 Oct 2019

PONE-D-19-24756R1 

Cost-effectiveness analysis of prostate-specific-antigen-based mass screening: Evidence from a randomised controlled trial combined with register data 

Dear Dr. Booth:

I am pleased to inform you that your manuscript has been deemed suitable for publication in PLOS ONE. Congratulations! Your manuscript is now with our production department. 

With kind regards,

on behalf of

Dr. Christopher J.D. Wallis 

Academic Editor

PLOS ONE